# Social Return on Investment of Social Prescribing via a Diabetes Technician for Preventing Type 2 Diabetes Progression

**DOI:** 10.3390/ijerph20126074

**Published:** 2023-06-07

**Authors:** Adam Skinner, Ned Hartfiel, Mary Lynch, Aled Wyn Jones, Rhiannon Tudor Edwards

**Affiliations:** 1Centre for Health Economics and Medicines Evaluation, Bangor University, Bangor LL57 2PZ, UK; 2Faculty of Nursing and Midwifery, Royal College of Surgeons in Ireland, D02 YN77 Dublin, Ireland

**Keywords:** Type 2 Diabetes Mellitus (T2DM), prediabetes, social return on investment (SROI), social prescribing (SP), physical activity (PA), overall health

## Abstract

In Wales, the prevalence of Type 2 Diabetes Mellitus (T2DM) has increased from 7.3% in 2016 to 8% in 2020, creating a major concern for the National Health Service (NHS). Social prescribing (SP) has been found to decrease T2DM prevalence and improve wellbeing. The MY LIFE programme, a scheme evaluated between June 2021 and February 2022 in the Conwy West Primary Care Cluster, aimed to prevent T2DM by referring prediabetic patients with a BMI of ≥30 to a diabetes technician (DT), who then signposted patients to community-based SP programmes, such as the National Exercise Referral Scheme (NERS), KindEating, and Slimming World. Although some patients engaged with SP, others chose to connect only with the DT. A Social Return on Investment (SROI) analysis was conducted to evaluate those patients who engaged with the DT plus SP, and those who connected solely with the DT. Relevant participant outcomes included ‘mental wellbeing’ and ‘good overall health’, which were measured at baseline (*n* = 54) and at the eight-week follow-up (*n* = 24). The estimated social value for every GBP 1 invested for participants who engaged with the ‘DT only’ ranged from GBP 4.67 to 4.70. The social value for participants who engaged with the ‘DT plus SP programme’ ranged from GBP 4.23 to 5.07. The results indicated that most of the social value generated was associated with connecting with the DT.

## 1. Introduction

Type 2 Diabetes Mellitus (T2DM) is a serious yet preventable health condition often caused by poor lifestyle choices. T2DM occurs when the pancreas is unable to produce sufficient amounts of insulin to mediate the higher levels of glucose entering the bloodstream. Impaired insulin production can lead to the development of other chronic conditions such as heart disease, stroke, blindness, and kidney failure [1,2,3]. Those considered at a high risk of developing T2DM are typically diagnosed with prediabetes [4,5,6]. Prediabetes is diagnosed using the measurement of HbA1c, with values between 42 to 47 mmol/mol (6.0 to 6.4 mmol/L) indicating prediabetes [7,8].

Diabetes has become a major concern in the United Kingdom (UK). In 2012, GBP 15.1 bn was spent on T2DM, with the costs predicted to reach GBP 39.8 bn by 2035/36 [9]. Preventable complications of T2DM also have significant financial consequences; for example, annual hospital costs relating to adverse events in diabetics range from GBP 1523 for transient ischemic attacks to GBP 20,954 for end-stage renal disease [10]. Conversely, patients achieving targets for HbA1c, cholesterol, and blood pressure led to significant annual healthcare cost savings, ranging from GBP 859 to 1037 per patient [11]. Thus, the prevention of diabetes and related complications is paramount.

This is especially true in Wales. In 2016, the prevalence of diabetes among Welsh residents aged 17 and older was 7.3%. By 2020, this figure had increased to 8%, the highest prevalence among the four nations of the UK [12,13]. NHS Wales spends approximately 10% of its annual budget (an estimated GBP 500 million) on the diagnosis and treatment of diabetes [13]. In response, in 2016, the Welsh Government formed the Diabetes Delivery Plan, a long-term strategy for both the treatment of pre-existing diabetes and the prevention of diabetes in the general population. Lifestyle interventions were highlighted as a key preventive measure. Launched in 2022, the All-Wales Diabetes Prevention Programme (AWDPP) is one such lifestyle intervention which emphasises dietary advice and SP [14].

SP involves referring patients with prediabetes to non-clinical, community-based interventions via a link worker, also known as a community navigator or health advisor [15]. Promoting a person-centred approach, SP offers patients access to therapeutic activities within a safe environment, coupled with mentor support. Research suggests that SP can lead to numerous benefits, including increased confidence and productivity [16], reduced levels of depression and anxiety, [17,18], and a reduction in the number of general practitioner (GP) and accident and emergency (A&E) visits [19,20]. SP activities can also reduce diabetic causal factors, including reduced waist circumference and decreased body mass index (BMI) [21]. Such effects can be vital in the long-term prevention of T2DM.

The objective of this evaluation was to conduct a social return on investment (SROI) analysis of the MY LIFE programme, an innovative lifestyle intervention developed by the Conwy West Primary Care Cluster, one of the largest clusters in Wales, with 11 GP practices and a practice population of approximately 64,000 people. The MY LIFE programme aims to prevent diabetes, reduce obesity, promote physical activity and improve mental wellbeing (Figure 1).

An overview of SP programmes signposted by the diabetes technician (DT):NERS consists of two supervised physical activity sessions per week, lasting approximately 1-h. NERS is delivered by an exercise professional who provides support throughout a 16-week period. The activities involved are primarily exercise and fitness classes.KE is a 12-week programme delivered by a registered dietician which includes weekly or fortnightly weigh-ins to measure progress. The dietician provides advice on healthy weight loss, eating habits, goal setting, physical activity, meal planning, dining out, and food labels.SW is a 12-week programme delivered by a SW group consultant with a focus on weight management advice and guidance, telephone support, buddy systems, and online support.

The DT plays a crucial role in the MY LIFE programme. During the eight-week evaluation period, participants received information and advice regarding exercise and diet from the DT every two weeks (Table 1). Participants received a catch-up call (15–20 min) with the DT at weeks two, four and six, obtaining advice on diet and physical activity, and referral to online educational materials and video content, which was especially relevant when pandemic restrictions limited attendance at in-person SP activities. Catch-up calls helped to determine how participants were engaging with the MY LIFE programme.

## 2. Materials and Methods

The SROI analysis compared the cost of implementing the MY LIFE programme with the social value generated. SROI is a type of social cost–benefit analysis (social CBA) [22]. Social CBA is recommended by the HM Treasury Green Book to assess interventions and their effects on wellbeing [23]. SROI uses the outcomes relevant to stakeholders and assigns monetary values to those outcomes. Examples of outcomes for participants in the MY LIFE programme were ‘mental wellbeing’ and ‘good overall health’.

Mental wellbeing was assessed using the Short Warwick–Edinburgh Mental Wellbeing Scale (SWEMWBS), a 7-item questionnaire used to assess the mental wellbeing of members within a population [24]. Good overall health was measured using the EuroQol EQ5D-5L questionnaire, a 5-item questionnaire to assess mobility, self-care, usual activities, pain/discomfort, and anxiety/depression [25].

After the quantity of outcomes was determined, outcomes were then monetised using the HACT Social Value Bank (SVB), which uses wellbeing valuation to estimate social value [26]. Wellbeing valuation offers a consistent and robust method for estimating the monetary value of relevant and material outcomes that often do not have market values. Wellbeing valuation was applied using two social value calculators: the social value calculator derived from the SVB, and the mental health social value calculator derived from SWEMWBS. In this study, the social value calculator was used to monetise the outcome of good overall health [26] with values assigned only to those participants who improved by a score of 0.05 or more on the EQ5D-5L utility index. A change of 0.05 or more in the utility index is considered ‘clinically relevant’ [27]. The mental health social value calculator was used to monetise ‘mental wellbeing’ based on the individual SWEMWBS scores at baseline and eight-week follow-up [28].

SROI evaluation involves five main stages: (1) identifying stakeholders, (2) developing a theory of change, (3) calculating inputs, (4) evidencing and valuing outcomes, and (5) estimating SROI ratios [22].

Ethical and governance approval for this study was granted by the NHS Integrated Research Application System (IRAS) in July 2021 (IRAS ID: 300887).

## 3. Results

### 3.1. Identifying Stakeholders

The primary stakeholders in this evaluation were patients with a diagnosis of prediabetes and a BMI score of ≥30 who participated in the MY LIFE programme. The NHS was also a key stakeholder, as participation in the MY LIFE programme was designed to reduce the demand for NHS health services.

### 3.2. Theory of Change

After the main stakeholders were identified, a theory of change was developed to illustrate the relationship between inputs, outputs, outcomes, and impact (Figure 2).

### 3.3. Calculating Inputs

Two main cost categories were identified: costs related to the DT and costs related to the delivery of SP. Costs related to the DT included a laptop, mobile phone, mobile phone contract, and the salary of the DT (30 h per week at GBP 10.40 per hour). The SP delivery costs for KindEating (KE) and Slimming World (SW) were provided by the lead dietician of the Conwy West Primary Care Cluster; delivery costs for NERS were provided by the fitness development manager at Conwy County Borough Council (Table 2).

### 3.4. Evidencing and Valuing Outcomes

Some 54 MY LIFE participants completed baseline questionnaires, and 24 participants completed the eight-week follow-up questionnaire. Data from questionnaires were used to gather information on participant health status, health service use, and outcomes related to mental wellbeing and good overall health. Data were analysed to determine the number of participants who improved, worsened, or experienced no change for each outcome. Baseline and eight-week follow-up scores were compared to identify changes in mental wellbeing (SWEMWBS) and good overall health (EQ5D-5L) (Table 3). Of the 24 participants who completed the baseline and follow-up questionnaires, 12 participants engaged with the diabetes technician only (DTO), and 12 participants engaged with the DT plus an SP activity (DT + SP). The results showed that the group who engaged with SP were slightly younger, and included more women than the group who chose to engage with DT only (Table 3).

#### 3.4.1. Good Overall Health

Improved good overall health was reported by participants who engaged with the DTO and DT + SP. Two of the twelve in the DTO group and three of the twelve in the DT + SP group reported clinically relevant improvements of 0.05 or more in their EQ5D-5L results.

#### 3.4.2. Wellbeing Valuation Using the Social Value Calculator

The HACT Social Value Calculator assigns a value of GBP 20,141 per year for good overall health. This monetary value is awarded only to those participants who experienced a change of 0.05% or more on the EQ5D-5L utility index from baseline to eight-week follow-up. Participants whose scores decreased by 0.05 or more (*n* = 0) would have been assigned a social value decrease of GBP 20,141 per year.

For participants in the DTO group, the total social value was GBP 40,282 for the two participants who experienced a gain of 0.05 or more on the utility index (Table 4). For participants in the DT + SP group, the total social value was GBP 60,423 for the three participants who reported a gain of 0.05 or more (Table 4).

#### 3.4.3. Deadweight, Attribution and Displacement

To avoid over-claiming, it is standard procedure in SROI analysis using the Social Value Calculator to consider deadweight, attribution, and displacement [22] (Table 5). The eight-week follow-up questionnaire indicated that the mean deadweight percentage was 43%, meaning 43% of improvements would have happened anyway, even without the intervention. The attribution percentage was 72%, suggesting that 72% of the change was due to the MY LIFE programme. The displacement percentage was 0%, meaning that the MY LIFE programme did not displace any other activities that would have improved health outcomes for participants (Table 5).

#### 3.4.4. Wellbeing Valuation Using Mental Health Social Value Calculator

Applying the HACT Mental Health Social Value Calculator, baseline and eight-week follow-up scores for SWEMWBS were quantified, and monetary values assigned to each participant [28]. A deadweight of 27% was subtracted [29], and the total social value was calculated for each participant (Table 6).

#### 3.4.5. Health Service Resource Use

Baseline and follow-up questionnaires asked participants about the number of visits they had with NHS services in the two months preceding the MY LIFE programme and in the two months during the MY LIFE programme. The total annual cost saving from reduced health service resource use for participants engaged with the DT + SP activity was GBP 138 per participant (Table 7) and GBP 167 for participants who engaged with the DTO (Table 8).

### 3.5. Calculating the SROI Ratios

The results indicated that for every GBP 1 invested in DT + SP activity, a social value of GBP 4.67 to 5.07 was generated per participant (Table 9). The social value for DTO participants ranged from GBP 4.23 to 5.07 per participant for every GBP 1 invested (Table 10).

## 4. Discussion

The SROI analysis showed that the DT plays a key role in generating social value for prediabetic participants, with SROI ratios ranging from GBP 4.67 to 4.70 for every GBP 1 invested for DT + SP activity, and from GBP 4.23 to 5.07 for the DTO. The findings also indicated that both groups of MY LIFE participants showed a reduced frequency of NHS health service resource use at the eight-week follow-up, with the exception of DTO participants who reported an increase of one outpatient visit.

The results suggest that positive social value outcomes were mainly a result of contact with the DT. It was estimated that between 54% (using the Social Value Calculator) and 67% (using the Mental Health Social Value Calculator) of the social value awarded to participants in the MY LIFE programme could be attributed to engagement with the DT, who provided telephone support and motivation to participants every two weeks during the eight-week intervention. Such a result highlights the importance of the DT in supporting prediabetic patients and preventing disease progression.

Although participants who engaged with the DT + SP experienced 33% to 46% more social value than the DTO group, the increased cost associated with the delivery of SP resulted in similar SROI ratios for both groups. The results in this study were undoubtedly affected by COVID-19 restrictions, which may have reduced the number of MY LIFE participants enrolled in SP activities. Some 50% of participants who completed baseline and follow-up questionnaires chose not to engage with SP, and a greater percentage of these participants were over the age of 50. Older people with prediabetes were likely to have been more hesitant to attend in-person SP activities during the pandemic.

### 4.1. Strengths

While previous UK studies have investigated the effects of SP activities for weight loss and reductions in diabetic symptoms [31,32], this is the first study to use an SROI methodology for evaluating SP for prediabetic patients. Furthermore, this study applied a consistent and robust methodology recommended by the UK Treasury—that of wellbeing valuation—using two different social value calculators.

### 4.2. Limitations

The study design lacked randomisation and a control group. Therefore, no comparisons could be made between a group that received an intervention (DTO or DT + SP) and a group that received no intervention [33]. However, this issue was mitigated by the inclusion of a follow-up questionnaire that measured deadweight, attribution, and displacement (Table 5).

This study also had a small sample size (*n* = 24), which may have led to increased variability and a decreased likelihood that the results reflected those of the general population of prediabetic patients [34]. In addition, the small numbers of participants in each of the SP programmes makes it difficult to determine if any of the observed differences in patient outcomes among the SP programmes were actually meaningful.

The participant retention rate for this study was low (44%). Although this percentage is less than half of the initial sample enrolled at baseline (*n* = 54), it is within the average range of retention rates (35% to 96%) for group-based weight management programmes [35]. Nevertheless, the low retention rate may have led to attrition bias.

Participant adherence to SP activity was not recorded. The DT was unable to determine how many SP sessions were attended by participants. It was therefore not possible to determine the number of participants who fully attended an SP programme, and the dosage needed to produce a positive effect.

Finally, this study took place during the COVID-19 pandemic, which caused a delay in referrals to SP and a reduction in SP uptake. Due to these circumstances, attendance at SP programmes decreased during the pandemic, potentially resulting in less social value attributed to participants who engaged with the DT + SP [36].

## 5. Conclusions

This research showed that the role of the DT was key in generating a positive SROI ratio. Regular contact with the DT and referral to SP led to improvements in good overall health and mental wellbeing for prediabetic patients. The results indicated that participant utilisation of NHS resources was reduced after participation in the MY LIFE programme.

Although the results of this study appear promising, there were important limitations, such as a lack of randomisation, a small sample size, the use of only one research site (Conwy West), insufficient monitoring of SP attendance, and reductions in availability of SP due to COVID-19 restrictions. The results showed that the total social value generated was greater for participants who engaged with a DT + SP activity. However, this greater social value did not ultimately provide higher SROI ratios due to the costs involved with delivering SP programmes. Future healthcare policy should support the role of the DT, while continuing to measure the effect of SP for prediabetic patients at a time when COVID restrictions are not in place.

## Figures and Tables

**Figure 1 ijerph-20-06074-f001:**
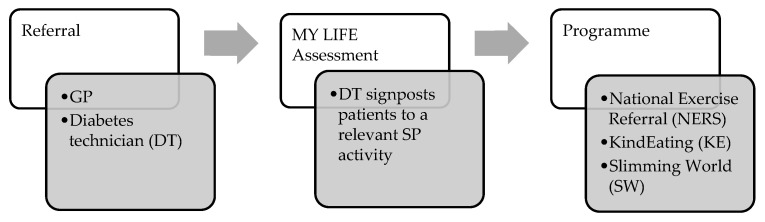
MY LIFE referral process.

**Figure 2 ijerph-20-06074-f002:**
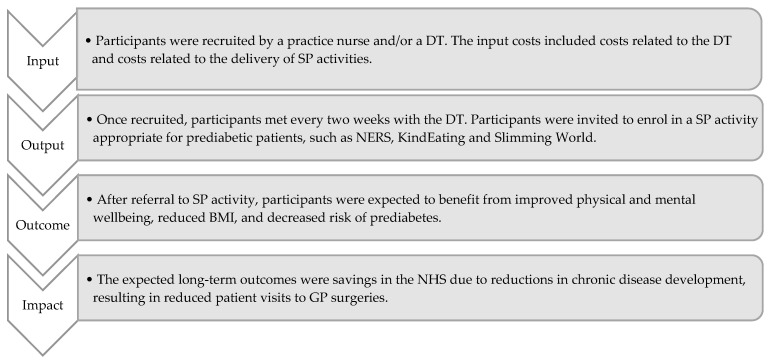
Theory of change.

**Table 1 ijerph-20-06074-t001:** Role of the DT during the eight-week MY LIFE programme.

Baseline and Follow-Up	
Baseline	DT contacts participant for a 30–40 min introductory session DT and participant discuss lifestyle, diet and physical activity DT signposts participant to SP activities in the community
Weeks 2–6	DT phones participant (15–20 min) every two weeks DT provides participant with advice on diet and physical activity DT suggests online content to improve physical activity and mental wellbeing
Week 8	DT contacts participant for a 30–40 min final session

**Table 2 ijerph-20-06074-t002:** Total costs per year for the MY LIFE programme.

Cost Category	Annual Costs per Participant (*n* = 54)
DT equipment costs	Mobile phone = GBP 60 (GBP 5 per month) GBP 60/54 patients = GBP 1.11 Laptop = GBP 947 GBP 947/5-year lifetime = GBP 189.40 GBP 189.40/54 patients = GBP 3.51 Total equipment costs = GBP 4.62
DT salary costs	GBP 10.40 per hr × 7.5 h per day = GBP 78 GBP 78 × 4 days per wk = GBP 312 GBP 312 × 52 wks per year = GBP 16,224 GBP 16,224/54 participants per yr Total salary costs = GBP 300
Total DT costs	GBP 304.61
SP delivery costs	GBP 258 (NERS) × 2 participants= GBP 516 GBP 75 (SW) × 2 participants = GBP 150 GBP 135.70 (KE) × 8 participants = GBP 1085.60 Total SP delivery costs = GBP 1751.60
Total SP delivery costs	GBP 145.96
Total Costs	GBP 450.57

**Table 3 ijerph-20-06074-t003:** Demographic overview of complete cases for MY LIFE participants (*n* = 24).

Category	DTO (*n* = 12)	DT + SP (*n* = 12)
Age	50% aged 50 and over	29% aged 50 and over
Gender percentage	50% Female	80% Female
Ethnic origin	100% White British	100% White British
Average SWEMWBS score at baseline	24.4	25.6
Average SWEMWBS score at 8 weeks	26	28.2
Average EQ5D-5L at baseline	0.801	0.823
Average EQ5D-5L at 8 weeks	0.803	0.845

**Table 4 ijerph-20-06074-t004:** Quantity of outcomes and total social value.

Outcomes (*n* = 24)	Indicators	Net Quantity	Financial Value	Total Social Value	Social Value per Participant
DT + SP Good Overall Health	EQ5D-5L	3/12	GBP 20,141 per year for good overall health	GBP 60,423	GBP 5035 (*n* = 12)
DTO Good Overall Health	EQ5D-5L	2/12	GBP 20,141 per year for good overall health	GBP 40,282	GBP 3357 (*n* = 12)

**Table 5 ijerph-20-06074-t005:** Deadweight, attribute, and displacement.

Outcomes	Total Social Value per Participant	Deadweight	Attribution	Displacement	Total Social Value per Participant
DT + SP	GBP 5035	43% (×0.57)	28% (×0.72)	0%	GBP 2066
DTO:	GBP 3357	43% (×0.57)	28% (×0.72)	0%	GBP 1378

**Table 6 ijerph-20-06074-t006:** Mental health Social Value Calculator.

Programme	ID	Age ^1^	Baseline	Week 8	GBP Value Baseline	GBP Value Week 8	Value Change	Value—27% Deadweight
NERS	1005	25–49	30	30	GBP 25,470	GBP 25,470	GBP 0	GBP 0
NERS	109	25–49	26	28	GBP 24,144	GBP 25,145	GBP 1001	GBP 731
KE	1014	50+	25	26	GBP 23,295	GBP 23,295	GBP 0	GBP 0.00
KE	817	50+	29	28	GBP 24,480	GBP 23,563	−GBP 917	−GBP 669
KE	909	25–49	18	21	GBP 10,523	GBP 20,831	GBP 10,308	GBP 7525
KE	107	25–49	16	19	GBP 8587	GBP 16,701	GBP 8114	GBP 5923
KE	2001	50+	28	27	GBP 23,563	GBP 23,563	GBP 0	GBP 0
KE	2011	50+	24	26	GBP 21,434	GBP 23,295	GBP 1861	GBP 1359
KE	715	25–49	26	28	GBP 24,144	GBP 25,145	GBP 1001	GBP 731
KE	8	25–49	18	24	GBP 10,523	GBP 23,383	GBP 12,860	GBP 9388
SW	903	25–49	31	33	GBP 25,811	GBP 25,811	GBP 0	GBP 0
SW	813	50+	22	22	GBP 19,947	GBP 19,947	GBP 0	GBP 0
DTO	401	50+	24	26	GBP 21,434	GBP 23,295	GBP 1861	GBP 1359
DTO	802	25–49	25	27	GBP 24,144	GBP 25,145	GBP 1001	GBP 731
DTO	803	25–49	21	25	GBP 20,831	GBP 24,144	GBP 3313	GBP 2418
DTO	906	50+	28	31	GBP 23,563	GBP 25,132	GBP 1569	GBP 1145
DTO	907	50+	31	33	GBP 25,132	GBP 25,609	GBP 477	GBP 348
DTO	911	25–49	28	30	GBP 25,145	GBP 25,470	GBP 325	GBP 237
DTO	101	50+	28	32	GBP 23,563	GBP 25,811	GBP 2248	GBP 1145
DTO	2006	50+	19	20	GBP 16,653	GBP 16,653	GBP 0.00	GBP 0
DTO	701	25–49	26	29	GBP 24,144	GBP 25,470	GBP 1326	GBP 968
DTO	713	50+	28	28	GBP 23,563	GBP 23,563	GBP 0	GBP 0
DTO	318	25–49	29	33	GBP 25,470	GBP 25,811	GBP 341	GBP 249
DTO	15	25–49	20	24	GBP 16,701	GBP 23,383	GBP 6682	GBP 4878
Total Social Value Per Participant enrolled with DT + SP								GBP 2082
Total Social Value Per Participant enrolled with DTO								GBP 1123

^1^ Mental health Social Value Calculator providing different monetary values for the same SWEMWBS score depending on the age category.

**Table 7 ijerph-20-06074-t007:** Health service resource use for DT + SP.

Service Use between Baseline and 8 Weeks	2 Months before Programme	2 Months during Programme	Difference in Visits	Cost per Visit ^1^	Cost Saving per 2 Months	Cost Saving per 12 Months
GP visits	7	3	4	GBP 39/visit	GBP 156	GBP 936
Nurse	6	6	0	GBP 44/visit	GBP 0	GBP 0
Outpatient	2	1	1	GBP 120/visit	GBP 120	GBP 720
999 Ambulance	0	0	0	GBP 231/visit	GBP 0	GBP 0
A&E ^2^	0	0	0	GBP 135/visit	GBP 0	GBP 0
Total cost saving					GBP 276	GBP 1656
Total cost saving per participant at 8 weeks (*n* = 12)						GBP 138

^1^ National Cost Collection: 2020-21—NHS Trust and NHS Foundation Trusts [30]. ^2^ Accident and Emergency department.

**Table 8 ijerph-20-06074-t008:** Health service resource use for DTO.

Service Use between Baseline and 8 Weeks	2 Months before Programme	2 Months during Programme	Difference in Visits	Cost per Visit ^1^	Cost Saving per 2 Months	Cost Saving per 12 Months
GP visits	8	2	6	GBP 39/visit	GBP 234	GBP 1404
Nurse	9	4	5	GBP 44/visit	GBP 220	GBP 1320
Outpatient	2	3	1	GBP 120/visit	−GBP 120	−GBP 720
999 Ambulance	0	0	0	GBP 231/visit	GBP 0	GBP 0
A&E	0	0	0	GBP 135/visit	GBP 0	GBP 0
Total cost saving					GBP 334	GBP 2004
Total cost saving per participant at 8 weeks (*n* = 12)						GBP 167

^1^ National Cost Collection: 2020-21—NHS Trust and NHS Foundation Trusts [30].

**Table 9 ijerph-20-06074-t009:** SROI ratio for DT + SP.

	SROI Ratio (Social Value Calculator)	SROI Ratio (Mental Health Social Value Calculator)
Total social value per participant	GBP 2066	GBP 2082
NHS cost savings per participant	GBP 138	GBP 138
Total financial value per participant	GBP 2204	GBP 2220
Total cost per participant	GBP 472	GBP 472
SROI ratio	GBP 4.67:GBP 1	GBP 4.70:GBP 1

**Table 10 ijerph-20-06074-t010:** SROI ratio for DTO.

	SROI Ratio (Social Value Calculator)	SROI Ratio (Mental Health Social Value Calculator)
Total social value per participant	GBP 1378	GBP 1123
NHS cost savings per participant	GBP 167	GBP 167
Total financial value per participant	GBP 1545	GBP 1290
Total cost per participant	GBP 305	GBP 305
SROI ratio	GBP 5.07:GBP 1	GBP 4.23:GBP 1

## Data Availability

The data presented in this study are available on request from the corresponding author.

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
