# Peer review of "Social Return on Investment of Social Prescribing via a Diabetes Technician for Preventing Type 2 Diabetes Progression"

_ijerph, 2023, doi:10.3390/ijerph20126074_

Round 1
Reviewer 1 Report
Referee report for ijerph-2290568 “Social return on investment analysis of social prescribing for preventing type 2 diabetes mellitus progression in prediabetic patients in Wales”
Summary
The paper describes the costs and (monetised) health benefits of using a diabetics technician (DT) and social prescribing (SP) for individuals at risk of type 2 diabetes. It presents results for a DT only approach and the combination of DT and SP.
Evaluation
SP is increasingly considered by policy makers and understanding how much (additional) value it provides to society is interesting. However, I have a few concerns about how much we can learn from the given study and suggest that the authors are more specific about this issue.
Main issues
1. The effect of the significant attrition of the sample makes the results hard to interpret, as there is no control group. For example, the SP could allow to retain individuals with worse health trajectories, which could make the comparison of DT and DT+SP flip.
2. I would prefer to observe results presented as total costs instead of costs per patient (given that we actually do not know anything about most patients – they left the sample). I do not think that will change anything in the relative results but would be clearer about what is measured.
3. The authors should be clearer that the effect of the SP they describe is a marginal effect of adding the SP to the DT. Currently all the benefit is attributed to the DT, but a SP alone may also have a large effect.
4. On p.5, the authors report the percentage of participants observing a 10% improvement.
a. First, using percentages in a sample of 12 participants appears misleading, as it suggests some level of fine-grained precision that is simply not there.
b. A 10% improvement appears an arbitrary cut-off. I would suggest to present the results in the DT and DT+SP arms in the same way as done in Table 3. Maybe just adding two further columns to that table is clearer that the current on-text description.
5. I would like the limitations section to be extended.
a. The paper currently mentions that there is no control group. However, I see the main issue in the combination of no control group and significant attrition, making it difficult to even just compare the DT and DT+SP groups. This should be added.
b. The paper discussed the small sample. The small sample also implies that statistical tests do not allow to investigate if any observed differences between the different programs are actually meaningful. Indeed, the authors do not claim that they can make any such statements, but it should be made more explicit in the limitations section.
